Variable effects of temperature on insect herbivory

Lemoine Nathan P. 1 lemoine.nathan@gmail.com
Burkepile Deron E. 1
Parker John D. 2
1 Department of Biological Sciences, Florida International University , Miami, FL , United States
2 Smithsonian Environmental Research Center, Smithsonian Institution , Edgewater, MD , United States
Singer Michael
Electronic publication date: 2014 May 6
Publication date: 2014
Volume: 2
Electronic Location ID: e376
Received 2014 Feb 7; Accepted 2014 Apr 22
Copyright: © 2014 Lemoine et al.
Copyright year: 2014
Copyright holder: Lemoine et al.
License: This is an open access article, free of all copyright, made available under the Creative Commons Public Domain Dedication. This work may be freely reproduced, distributed, transmitted, modified, built upon, or otherwise used by anyone for any lawful purpose.
License URL: https://creativecommons.org/publicdomain/zero/1.0/

Keywords: Climate change, Hierarchical model, Thermal response curve, Lepidoptera, Coleoptera, Hymenoptera

Funding: FIU Work was supported by a Presidential Fellowship from FIU (NL). The funders had no role in study design, data collection and analysis, decision to publish, or preparation of the manuscript.

==============================
Rising temperatures can influence the top-down control of plant biomass by increasing herbivore metabolic demands. Unfortunately, we know relatively little about the effects of temperature on herbivory rates for most insect herbivores in a given community. Evolutionary history, adaptation to local environments, and dietary factors may lead to variable thermal response curves across different species. Here we characterized the effect of temperature on herbivory rates for 21 herbivore-plant pairs, encompassing 14 herbivore and 12 plant species. We show that overall consumption rates increase with temperature between 20 and 30 °C but do not increase further with increasing temperature. However, there is substantial variation in thermal responses among individual herbivore-plant pairs at the highest temperatures. Over one third of the herbivore-plant pairs showed declining consumption rates at high temperatures, while an approximately equal number showed increasing consumption rates. Such variation existed even within herbivore species, as some species exhibited idiosyncratic thermal response curves on different host plants. Thus, rising temperatures, particularly with respect to climate change, may have highly variable effects on plant-herbivore interactions and, ultimately, top-down control of plant biomass.

Introduction

Environmental temperature drives a number of important ecological interactions, including competition, predation, and herbivory, by determining the metabolic rates of ectothermic organisms (Vassuer & McCann, 2005; O’Connor et al., 2009; Vucic-Pestic et al., 2011). As metabolic demands increase exponentially with rising temperatures, consumers generally either increase food intake or switch to higher quality diets to offset the rising costs of metabolism (O’Connor, 2009; Lemoine et al., 2013). As a result, predation and herbivory rates tend to increase exponentially with increased temperature (Hillebrand et al., 2009; Vucic-Pestic et al., 2011). However, both consumption rates and fitness can decline precipitously once a species encounters temperatures beyond its thermal optimum (Lemoine & Burkepile, 2012). A suite of adaptive and evolutionary factors determine these thermal optima, such that a given consumer community may contain species with considerably different thermal response curves (Angilletta, Steury & Sears, 2004; Angilletta, 2009). To date, however, few (if any) studies have examined variation in thermal response curves for a community of co-occurring herbivores (Buckley, Nufio & Kingsolver, 2014).

Insect herbivores can be an important top-down force in terrestrial ecosystems, controlling plant biomass (Carson & Root, 2000), maintaining species diversity (Bagchi et al., 2014), reorganizing competitive hierarchies (Kim, Underwood & Inouye, 2013), and enhancing nutrient cycling (Metcalf et al., 2014). Because insects are ectothermic, their physiological rates, including consumption and growth rates, are directly tied to environmental temperature. By extension, top-down control of plant biomass is also likely to be temperature dependent. Theoretical and experimental studies suggest that herbivory rates should increase exponentially with rising temperatures (O’Connor, Gilbert & Brown, 2011). One potential shortcoming of theoretical examinations of temperature-driven herbivore-plant interactions is that they do not often incorporate variation in thermal response curves within an assemblage of herbivore species. In part, this is because thermal response curves for multiple herbivore species on a single host plant remain mostly uncharacterized. Species are often examined singly, and ecologists have focused on a few readily available model species like Spodoptera spp. (Stamp & Yang, 1996) and Manduca sexta (Kingsolver & Woods, 1998). In contrast, the herbivore guild within a community, or even on a single host plant, can vary from 1–100s of species, each with different life histories, climatic niches, evolutionary histories, and dietary needs that may drive vastly different thermal response curves (Buckley, Nufio & Kingsolver, 2014).

Multiple factors aside from evolutionary history and local adaptation can determine the shape of a species’ thermal response curve. For example, plant chemical defenses can become more or less effective at high temperatures depending on the identity of the herbivore, plant, and chemical compounds in question (Stamp & Osier, 1998; Stamp, Yang & Osier, 1997). Similarly, different insect species can become more or less nutrient-limited at higher temperatures, which is also contingent on host plant quality (Kingsolver et al., 2006; Kingsolver & Woods, 1998; Lemoine et al., 2013). Thermal response curves therefore likely differ among herbivore species and within a single herbivore species utilizing different hosts. Given the interest in predicting the effects of climate change on trophic interactions and community structure (Singer, Travis & Johst, 2013; Urban, Tewksbury & Sheldon, 2012), we sought to determine whether temperature influences herbivory in a predictable manner based on a few easily measured variables of plant nutritional quality.

Here, we report thermal response curves of consumption rates for 21 herbivore-plant pairs, encompassing 14 herbivores and 12 plant species (Table 1). We asked two specific questions: (1) what is the extent of within- and among-species variation in thermal response curves of consumption for insect herbivores? and (2) can plant nutritional quality explain variation in thermal response curves? By working with multiple species of both herbivores and plants, we demonstrate that thermal response curves vary substantially both among and within herbivore species. However, we were unable to detect any influence of plant nutritional quality on the overall shape of the thermal response curve across taxa, suggesting that thermal response curves are idiosyncratic and highly variable among plant-herbivore pairs.

Table 1 Herbivores and plants used in this study.

Herbivore—plant pairings used in feeding assays. Species marked with (I) are introduced species; common names are given in parentheses. Below each species name, we have listed the order and family of each species.

Herbivore species	Herbivore diet	Plant species	
Unidentified tortricid
(Lepidoptera, tortricidae)	Specialist	Lindera benzoin (northern spicebush)	
Arge scapularis (elm argid sawfly)
(Hymenoptera, argidae)	Specialist	Ulmus rubra (slippery elm)	
Atteva aurea (ailanthus webworm)
(Lepidoptera, yponomeutidae)	Specialist	Ailanthus altissima (tree-of-heaven) (I)	
Chrysochus auratus (dogbane beetle)
(Coleoptera, chrysomelidae)	Specialist	Apocynum cannabinum (dogbane)	
Danaus plexippus (monarch butterfly)
(Lepidoptera, nymphalidae)	Specialist	Asclepias syriaca (common milkweed)	
Epimecis hortaria (tulip tree beauty)
(Lepidoptera, geometridae)	Generalist	Lindera benzoin (northern spicebush)
Liriodendron tulipifera (tulip poplar)
Sassafras albidum (sassafras)	
Euchaetes egle (milkweed tussock moth)
(Lepidoptera, arctiidae)	Specialist	Apocynum cannabinum (dogbane)
Asclepias syriaca (common milkweed)	
Hyphantria cunea (fall webworm)
(Lepidoptera, arctiidae)	Generalist	Acer negundo (box elder)
Liquidambar styraciflua (sweetgum)	
Malacosoma americanum (eastern tent caterpillar)
(Lepidoptera, lasiocampidae)	Generalist	Prunus serotina (black cherry)	
Melanophia canadaria (canadian melanophia)
(Lepidoptera, geometridae)	Generalist	Acer negundo (box elder)
Lindera benzoin (northern spicebush)
Sassafras albidum (sassafras)	
Nematus tibialis (locust sawfly)
(Hymenoptera, tenthredinidae)	Specialist	Robinia pseudoacacia (black locust)	
Papilio polyxenes (black swallowtail)
(Lepidoptera, papilionidae)	Specialist	Foeniculum vulgare (fennel)	
Papilio troilus (spicebush swallowtail)
(Lepidoptera, papilionidae)	Specialist	Lindera benzoin (northern spicebush)
Sassafras albidum (sassafras)	
Saucrobotys futilalis (dogbane webworm)
(Lepidoptera, crambidae)	Specialist	Apocynum cannabinum (dogbane)	

Methods

All experiments were conducted at the Smithsonian Environmental Research Center (SERC), in Edgewater, MD, USA from June–August 2012. Using laboratory feeding assays, we evaluated the feeding performance of 14 herbivore species from three Orders (Lepidoptera, Coleoptera, Hymenoptera) on 12 plant species (Fig. 1, Table 1). Herbivores were collected by hand from host plants in the forests and fields on the SERC premises throughout the summer. All herbivores were kept in a cage and fed leaves from the plant species on which they were collected. Individuals were used in feeding assays within 24 h of collection. No individual was used more than once. As herbivores were opportunistically collected, the number of replicates per host plant/temperature combination varied depending on the number of herbivores found (Table S1). Gregarious species (e.g., Hyphantria cuneata) have higher replicate numbers than do rare, non-gregarious species (e.g., Danaus plexippus). A single lepidopteran species could not be identified beyond the Tortricidae family. We focused on folivorous individuals, using larvae from lepidopteran and hymenopteran species and adults only from a single coleopteran species (Chrysochus auratus).

Figure 1 Chrysochus auratus.

Chrysochus auratus, the dogbane beetle, feeding on Apocynum cannabinum.

Feeding assays

In no-choice assays, a single individual was weighed and placed in a single rearing cup with a single, pre-weighed leaf from a potential host plant (see Table 1). Each rearing cup was randomly assigned to one of four temperatures (20, 25, 30, and 35 °C, see Table S2 for temperature and light data from each growth chamber) maintained in growth chambers on a 14:10 light:dark cycle. Temperatures were selected to represent a realistic set of temperatures during the spring in summer months. Data from a nearby NOAA weather station (Annapolis, MD) indicate that temperatures can range from 20–35° during the summer months (June–July, Fig. S1). Leaf petioles were placed in water-filled microcentrifuge tubes capped with cotton to prevent desiccation, and we observed no obvious differences in leaf turgor during the assays. After 24 h, herbivores and leaves were reweighed to estimate consumption rates. Feeding assays of this duration have been used to assess herbivore performance and dietary preferences in lepidopterans (Kingsolver & Woods, 1998; Kingsolver & Woods, 1997) and coleopterans (Gange et al., 2012; Lemoine et al., 2013). We divided all consumption rates by the initial mass of the individual used in the feeding assay to account for variation in body size among replicates. Consumption rates are reported as grams consumed per gram body mass.

Control assays with no herbivores accounted for autogenic change in leaf weight over the 24 h period (n = 5 per plant species per temperature). Leaves of all plant species except L. styraciflua gained mass over 24 h in the absence of herbivores. Larger leaves gained more mass than did smaller leaves. We therefore used species-specific equations to correct for autogenic change in leaf mass (Table S3) rather than using mean change in leaf mass across all autogenic controls. Mass-specific autogenic changes, whether positive or negative, were added to leaf final weights. Negative autogenic changes (i.e., plants lost mass in control assays) would therefore lower estimates of consumption and vice versa. In total, we conducted 552 no-choice feeding assays, resulting in 496 observations after removing individuals that died or molted overnight (final replicate numbers for each herbivore/plant/temperature combination given in Table S1).

Plant traits

To assess the mechanisms by which temperature affected herbivore performance among plant species, we quantified nutritional characteristics of undamaged leaves (n = 3–5) of each plant species, all collected from unique individuals. Prior to all nutrient content analyses, leaves were weighed, dried to a constant mass at 60 °C, and re-weighed to estimate water content. Dried leaf material was ground to a fine powder for carbon (C), nitrogen (N), and phosphorus (P) analyses. Percent C and N were estimated using an EAI CE-440 elemental analyzer (Exeter Analytics, Coventry, UK). Phosphorus content was determined using dry oxidation-acid hydrolysis extraction followed by colorimetric analysis on a microplate spectrophotometer (PowerWave XS; Biotek, Winooski, VT).

Statistical analyses

We used a Bayesian hierarchical model to determine thermal response curves of consumption for each herbivore-plant pairing. This allowed us to estimate parameters for the overall trend in consumption with increasing temperature, parameters for each herbivore-plant pairing, and the impact of plant nutritional quality on these parameters. A multilevel model is particularly appropriate for handling unbalanced data and small sample sizes for some herbivore-plant pairings, but some of the predicted responses for less well-sampled taxa will be pulled heavily towards the overall mean response (Gelman & Hill, 2007). Although there could be a phylogenetic signal in the patterns of thermal curves of different insect herbivores, we did not have sufficient replication within genera or families to address this question. Most species were in unique families (only three families had more than one species represented) and all but three species were lepidopterans (Table 1). Thus, we did not incorporate the possibility of a phylogenetic signal into our analyses.

Thermal reaction norms of consumption describe the influence of temperature on consumption rates. Regardless of the specific equation used to model a reaction norm, all thermal reaction norms are characterized by a thermal minimum below which consumption is zero, a thermal optimum where consumption rate is maximized and beyond which consumption declines, and a thermal maximum, above which consumption is zero. We modeled the thermal reaction norm of consumption rates for each herbivore-plant pairing using a quadratic exponential (i.e., Gaussian) curve because such curves often describe thermal reaction norms (Angilletta, 2006): yij=expaj+bjTempij+cjTempij2+εij

where yij is consumption of the ith observation in the jth herbivore-plant pair and εij is residual error. We assumed that errors were normally distributed with a constant variance, but the variance was allowed to differ for each curve due to differing numbers of replicates among herbivore-plant pairings. Hereafter, parameters will be referred to as the intercept (a), exponential (b), and Gaussian (c) terms. The intercept a denotes mean consumption rate (since all predictor variables were standardized, see below), the exponential term b denotes the rate at which consumption initially increases with temperature, and the Gaussian term c denotes the extent to which consumption rates level off or decline at high temperatures.

Plant nutritional quality can affect the shape of the thermal response curve by influencing any one of the three parameters that determine the shape of the Gaussian curve. Therefore, each parameter (intercept, exponential, and Gaussian) was modeled as function of nitrogen, phosphorus, and water content of the given plant for each herbivore-plant thermal response curve. For example, the exponential term of the jth curve was a linear function of plant quality: bj=μb+γ1Nj+γ2Pj+γ2H2Oj+δj

where μb is the overall, community-level linear parameter, and γ1, γ2, and γ3 represent the influence of nitrogen (%N), phosphorus (%P), and water content (%H2O) respectively on the exponential parameter of the jth thermal response curve. δj is a multivariate normal error term. Thus, mean consumption rate (a), the rate of increase with temperature (b), and the extent of curvature in the thermal response curve (c) were all modeled as linear functions of plant nutritional content. The random effects for each curve (i.e.,  parameters aj, bj, cj) were assumed to come from a multivariate normal distribution, allowing for covariance among parameter estimates.

All predictor variables were standardized prior to analysis to speed chain convergence. For all models, four MCMC chains were run for 5,000 ‘burn-in’ iterations to allow for chain convergence. Posterior distributions of each parameter were simulated by saving the 20th sample from an additional 5,000 posterior simulations, resulting in 1,000 independent estimates (250 per chain, with four chains). Chain convergence and autocorrelation were assessed using trace plots and density plots of posterior simulations. Each parameter was given a mildly uninformative prior normal distribution (N(0, 1)); variance parameters were given uninformative prior uniform distributions (U(0, 100)). Because predictors were standardized, the magnitude of parameter estimates will be small, such that a standard normal distribution is relatively uninformative. For each parameter, we calculated the 80% and 95% Bayesian credible interval (CI) from the posterior simulations. Parameters whose 95% CI excluded zero were considered highly significant, whereas parameters whose 80% CI excluded zero were considered marginally significant. If the 80% CI included zero, we assumed that the parameter had a low probability of being important. All assumptions of normality and homogenous variances were examined using residual plots. All analyses were conducted using Python v2.7. Bayesian models were evaluated using STAN v2.1 (Stan Development Team, 2013), accessed via PySTAN. Python code for the hierarchical model is available as Appendix S1. All code and raw data are available on the corresponding author’s website1 and will be uploaded to the Dryad database.2

Climate change simulations

We sought to understand how potential variability in thermal response curves among herbivores interacts with climate change to alter potential top-down control of plant biomass. Thus, we next built a simple model utilizing observed feeding rates and temperatures to estimate cumulative consumption first over one growing season, and then cumulative consumption given two climate change scenarios, +3 and a +5 °C increases in temperature, a moderate and severe climate warming scenario, respectively (IPCC, 2007). We obtained hourly temperature records for June–August 2013 from the NOAA weather station in Annapolis, MD (Fig. S2). We then used the 1000 posterior draws of observed feeding rates to estimate hourly consumption rates (including parameter uncertainty) for each herbivore-plant combination across the observed temperature range. Most hourly temperature readings were within the 20–35 °C range used in our experiments (Fig. S2). This yielded 1000 estimates of cumulative consumption for every herbivore-plant pair. We then simulated climate change by adding 3° and 5° to hourly temperature records and repeating the above calculations.

This method makes several important assumptions: (1) thermal effects of climate change can be approximated by adding a constant increase in temperature to all hourly temperature records, (2) that an individual feeds at a constant rate for the entire season with no variation as the instar grows over time, and (3) that a single individual is responsible for feeding, or multiple non-overlapping individuals immediately replace one another upon dying to maintain a constant consumption rate across the growing season. These assumptions will often not hold true so our method of assessing climate change is a relatively coarse picture of how climate change may affect herbivory rates over the course of an entire season.

Results

The exponential parameter (b) of overall consumption rates was significantly greater than zero, indicating that overall consumption rates did increase exponentially with temperature (Figs. 2 and 3). However, the increase was restricted to temperatures between 20 and 30 °C (Fig. 1). The 95% CI of the Gaussian parameter (c) narrowly included zero, but the bulk of the posterior distribution for this parameter lay below zero, indicating that overall consumption rates began to level off at temperatures above 30 °C (Pr(<0) = 0.96, Fig. 3). Accordingly, our model predicts relatively little change in overall consumption rates between 30 and 35 °C (Fig. 2). Variance in consumption rates among herbivore-plant pairs also increased substantially with rising temperature. At 20°, variance among mean herbivore-plant consumption rates was 0.45, while at 35° this variance increased to 1.43. Thus, variability in consumption rates among herbivore-plant pairs increased by over 300%. As a result, at 20 °C the predicted mean community-level consumption rates lie between 0.44 and 1.13 g per day (95% CI). Estimates of mean community-level consumption were more uncertain at higher temperatures, lying between 0.93 and 2.13 g per day (95% CI).

Figure 2 Effect of temperature on overall consumption rates.

Shaded area represents the 80% (blue) and 95% (green) credible interval of the prediction. Line shows the median posterior prediction.

Figure 3 Parameter estimates of overall consumption rates.

Points represent the median posterior estimate, while lines show the 80% (thick line) and 95% (thin line) CI.

Uncertainty regarding overall consumption rates at higher temperatures stems from idiosyncratic thermal response curves among herbivore-plant pairs (Figs. 4 and 5). Six herbivore-plant pairs (Arge scapularis–Ulmus rubra, Chrysochus auratus–Apocynum cannibinum, Hyphantria cunea–Acer negundo, H. cunea–Liquidambar styraciflua, Melanophia canadaria–Lindera benzoin, Papilio troilus–S. albidum) had Gaussian parameters (c) that were moderately or significantly different from zero, indicating decreasing consumption rates at higher temperatures (Figs. 4 and 5). An additional eleven herbivore-plant pairs (Atteva aurea–Ailanthus altissima, Danaus plexippus–Asclepias syriaca, Epimecis hortaria–L. benzoin, E. hortaria–S. albidum, Euchaetes egle–Asclepias syriaca, Malacosoma americanum–Prunus serotina, Melanophia canadaria–Acer negundo, Nematus tibialis–Robinia pseudoacacia, P. troilus–L. benzoin, Saucrobotys futilalis–A. cannibinum, Unidentified Tortricid – L. benzoin) increased consumption with warming throughout the entire temperature range, where the exponential parameter (b) was significantly or moderately different from zero. In some cases the parameter value was small enough that the fit was approximately linear (e.g., Epimecis hortaria–Sassafras albidum, Figs. 4 and 5). An additional four herbivore-plant combinations (Danaus plexippus–A. syriaca, E. hortaria–Liriodenron tulipifera, Euchaetes egle–Apocynum cannibinum, M. canadria–S. albidum, Papilio polyxense–Foeniculum vulgare) showed no detectable change in consumption rate with increasing temperature.

Figure 4 Effect of temperature on consumption rates for each herbivore-plant pair.

The thick line shows the median posterior estimate, shaded area shows the 95% confidence interval. Solid lines are significant at the 95% level, while dotted lines are significant at the 80% level. In some cases, a dotted line has a highly significant linear term but a moderately significant quadratic term (e.g., Hyphantrea cunea–Liquidambar styraciflua, see Fig. 4). Points depict mean consumption rates (±1 S.E.). Points are weighted by sample size, such that larger points contain more observations, to show how predictions for low sample sizes are pulled towards the overall response.

Figure 5 Parameter estimates for each herbivore-plant pair.

Points represent the median posterior estimate, while lines show the 80% (thick line) and 95% (thin line) CI.

Even within herbivore species, thermal response curves varied considerably. For example, E. hortaria consumption of L. benzoin, and to a lesser extent S. albidum, increased exponentially with temperature (Fig. 4). However, E. hortaria consumption of L. tulipifera did not vary significantly over the observed temperatures (Figs. 4 and 5). Likewise, P. troilus increased consumption of L. benzoin across temperatures, but consumption of S. albidum began to decline at 35° (Fig. 4). We were not able to detect any effect of plant nutritional content on the shape of thermal response curves among herbivore-plant pairs (Fig. 6).

Figure 6 Parameter estimates for nutritional content effects on thermal response curves.

Points represent the median posterior estimate, while lines show the 80% (thick line) and 95% CI.

High intra- and interspecific variability among thermal response curves for each herbivore-plant pair led to variable effects of increasing temperatures on potential top-down control of plant biomass (Fig. 7). For example, a 3 °C increase in temperatures resulted in less than a 20% increase in cumulative consumption for twelve herbivore-plant pairs compared to what is predicted for current temperatures, while leading to a >30% increase for four herbivore-plant pairs (Fig. 7). A 5 °C increase in temperature exacerbated this variability, as five herbivore-plant pairs exhibited <20% increase in cumulative consumption and four herbivore-plant pairs exhibited a >50% increase (Fig. 7). Further, increased warming from 3 to 5 °C had highly variable impacts on cumulative consumption rates among herbivore-plant pairs. For example, Atteva aurea nearly doubled its estimated consumption of Ailanthus altissima as warming increased from 3 to 5 °C, whereas estimated consumption by Melanophia canadaria was unaffected by temperature increases beyond 3 °C (Fig. 7). Thus, high intra- and interspecific variability in the herbivore-plant thermal response curves led to high variability in potential climate change effects on top-down control of plant biomass over an entire growing season.

Figure 7 Change in cumulative consumption due to warming for each herbivore-plant pair.

Percent change (±1 S.E.) in cumulative consumption resulting from a 3 and 5 °C increase in warming for all herbivore-plant pairs.

Discussion

Temperature influences herbivory rates via direct effects on insect herbivore physiology. However, it is currently unclear how temperature affects top-down control of plant biomass at the community or species level. Our data suggest that the influence of rising temperature on potential top-down control of plant biomass via herbivory depends upon the identity of the herbivore-plant pair under consideration. Such variability in the relationship between consumption rate and temperature will make it difficult to predict the effects of temperature changes, i.e., climate change, on top-down control of plant biomass.

Theory predicts that herbivory rates should increase exponentially with rising temperature more quickly than primary production, reducing standing plant biomass at higher temperatures (Gillooly et al., 2001; O’Connor et al., 2009; O’Connor, Gilbert & Brown, 2011) However, meta-analyses of thermal response curves report substantial variability among species. Indeed, approximately 40% of the thermal response curves examined by Dell, Pawar & Savage (2011) exhibited curvature, wherein the thermal response curve began to decrease at high temperatures. In our study, 33% of the herbivore-plant pairs exhibited substantial curvature, reducing consumption of plant biomass at high temperatures, thereby contradicting theoretical predictions of exponential increases in top-down control of plant biomass at high temperatures. Often, reduced consumption rates at high temperatures result from metabolic demand exceeding energetic supply, such that energy available for tasks beyond cellular maintenance, such as movement, feeding, or digestion, decreases sharply at high temperature (Somero, 2011). This results in rapid decreases in consumer fitness at temperatures beyond an organism’s thermal optimum (Lemoine & Burkepile, 2012). We show that community-level herbivory rates display the same, albeit much less pronounced, curvature as do some individual species. The slow decline at higher temperatures, rather than a rapid drop-off beyond some threshold value, is a result of species-specific variation in thermal response curves. Almost half (43%) of plant-herbivore pairings did not show signs of decreased consumption at higher temperature, while one showed evidence of declining consumption beyond 30 °C, leading to no net change in overall consumption rates at higher temperatures.

Such variation in thermal response curves makes predicting the effects of temperature changes (i.e., microhabitat variation, seasonal effects, climate change) on herbivore-plant interactions challenging in the absence of species-specific information. Indeed, a generalist herbivore may have as many thermal response curves as host plants (Lemoine et al., 2013) We report similar patterns here. For example, Epimecis hortaria, the tulip tree beauty moth, rapidly increased consumption of both Lindera benzoin and Sassafras albidum with warming, but the increased consumption of S. albidum was much slower. In contrast, E. hortaria showed no relationship between consumption of Liriodendron tulipifera and temperature. Similarly, Papilio troilus increased consumption of both L. benzoin and S. albidum with increasing temperature, but consumption of S. albidum began to decrease at 35 °C, and consumption of L. benzoin showed no curvature.

Given the high variation in thermal response curves among herbivore-plant combinations, predicting the effects of climate change on the top-down control of plant biomass remains challenging. Some studies have ascribed a single thermal response curve to herbivore species, demonstrating that plant biomass will decrease in a warming world as herbivory rates outpace primary production (O’Connor, Gilbert & Brown, 2011) Our results suggest that using a single consumption-temperature relationship for all herbivores can substantially overestimate the impact of climate change on plant biomass. For example, between 20 and 30 °C, both Chrysochus auratus and Saucrobotys futilalis increased consumption of Apocynum cannibinum. However, at 35° C. auratus decreased consumption while S. futilalis continued to increase consumption, resulting in little change in overall consumption rates on A. cannibinum beyond 30 °C.

When we integrated these changes in consumption over a full growing season, we showed that top-down control on plant biomass is likely to increase with increasing temperature but the magnitude of the increase depends on the herbivore-plant combination. Over the course of a summer, simulated warming resulted in >20% increase in cumulative consumption for 10 herbivore-plant pairs, just under half of the pairings examined. Conversely, simulated climate change resulted in >40% increase in cumulative consumption for five herbivore-plant pairs. Overall, the change in consumption ranged from no change to an increase of over 60%. This variability in consumption may explain why studies documenting significant effects of warming on top-down control of plant biomass typically examine one herbivore species (Chase, 1996; Barton, Beckerman & Schmitz, 2009) while studies focusing on entire herbivore communities report weak or negligible effects of warming (Richardson et al., 2002).

Surprisingly, we were unable to detect any influence of plant nutritional quality on the shape of thermal response curves. Based on previous work (Lemoine et al., 2013) we expected consumption rate to increase more rapidly on plants of higher nutritional quality. Conversely, compensatory feeding predicts that consumption rates should increase more rapidly with temperature for plants of low nutritional quality as herbivores attempt to fuel rising metabolic demands (e.g., Williams, Lincoln & Thomas, 1994). Our data suggest that plant nutritional content had little effect on thermal response curves among herbivore species. However, prior work has found that the relationship between temperature and consumption rates within a given species can vary with dietary quality. For example, the Japanese beetle Popillia japonica increased growth and consumption rates at high temperatures only on host plants with high nitrogen and carbon concentrations (Lemoine et al., 2013). This may also be the case in our data. Within an herbivore species, we can distinguish some patterns related to plant quality. For example, the generalist herbivore Epimecis hortaria increased consumption rapidly with warming only on higher nitrogen plant species. Within a plant species, however, patterns were less clear, as particular herbivore species were equally likely to have unimodal or exponential curves when feeding on the same plant. Thus, across all 21 plant-herbivore pairings, we were unable to detect an overall pattern relating plant quality to multiple thermal reaction norms. Thus, dietary quality may be more important for determining thermal response curves within herbivore species but cannot predict the shape of consumption thermal response curves among herbivore and plant species.

We focused on examining herbivore response to increasing temperature while holding plant phytochemistry constant, but rising temperatures might also affect plant phytochemistry. Plant growth rates increase with rising temperatures (Veteli et al., 2002), which could alter nutritional content or concentrations of defensive compounds as plants shuttle more resources into growth (Coley, Bryant & Chapin, 1985). However, studies have shown that the effects of temperature on plant secondary chemistry are highly idiosyncratic among species (Veteli et al., 2002; Zvereva & Kozlov, 2006). Furthermore, although variable temperature can alter nutritional quality within a species, variation caused by temperature is substantially lower than inherent variation among plant species (Aerts et al., 2009). However, the effects of rising temperature on plant chemistry must be considered more completely before applying results such as ours in a climate change context.

One potential caveat of our study is small sample size at many herbivore-plant-temperature combinations. Given that we used field-collected organisms, sample size varied considerably depending on the rarity of the species. Common and/or gregarious species, like E. hortaria, M. canadaria, and H. cunea, have much higher sample sizes than rare or cryptic species, like D. plexippus and A. scapularis. Thus, some herbivore-plant pairs show considerable variability in the estimated thermal response curve and, in some cases, the prediction was heavily influenced by the overall response. However, most work regarding the influence of temperature on herbivory and its interaction with diet quality focus on a few readily available lepidopteran herbivores (Kingsolver & Woods, 1998; Kingsolver et al., 2006). The influence of temperature on herbivory by the majority of the species reported here was heretofore unknown, and our research adds considerably to the body of work documenting the importance of temperature on rates of herbivory.

In summary, we show that herbivory, and therefore potential top-down control of plant biomass, is highly contingent upon environmental temperature. While theoretical predictions suggest that climate change might increase top-down control of plant biomass, our results indicate that the effects of temperature on herbivory rates are highly variable. A single plant species might experience more or less herbivory at higher temperatures, depending on the identity of the herbivores present. Insects often control plant community structure (Carson & Root, 2000) and dominance hierarchies among plant species (Kim, Underwood & Inouye, 2013) Thus, studies documenting the species-specific effects of temperature on insect herbivory levels will be crucial to understanding how climate change might affect community composition in the plant-herbivore assemblages of the future.

Supplemental Information

Figure S1 Daily average, minimum, and maximum temperatures in Annapolis, MD from April–August 2013

Click here for additional data file.

Figure S2 Hourly temperature measurements from June–August 2013 from the NOAA weather station at the US Naval Academy, Annapolis, MD

Click here for additional data file.

Table S1 Number of replicates for each plant-herbivore pair used in this study

Click here for additional data file.

Table S2 Growth chamber temperature and light data

Temperature data for each growth chamber, collected by HOBO pendant temperature loggers (HOBO UA-002 pendant loggers, Onset Computer Corporation, Bourne MA).

Click here for additional data file.

Table S3 Mass-correction equations

Regression equations used to correct for autogenic change for each leaf species, where y is the correction factor and Massi is the initial leaf mass. The response variable is change in leaf weight in the absence of herbivory.

Click here for additional data file.

Appendix S1 Model code

Model code used for all analysis and graphs using Python v2.7 and PySTAN v2.1

Click here for additional data file.

We would like to thank W Drews, J Shue, S Cook-Patton, and A Shantz for help collecting insects, running feeding assays, and insights on the manuscript.

Additional Information and Declarations

Competing Interests

Author Contributions

Data Deposition

1 http://www.natelemoine.com

2 http://datadryad.org/

John D. Parker is an employee of the Smithsonian Environmental Research Center.

Nathan P. Lemoine conceived and designed the experiments, performed the experiments, analyzed the data, contributed reagents/materials/analysis tools, wrote the paper, prepared figures and/or tables, reviewed drafts of the paper.

Deron E. Burkepile conceived and designed the experiments, wrote the paper, reviewed drafts of the paper.

John D. Parker conceived and designed the experiments, contributed reagents/materials/analysis tools, wrote the paper, reviewed drafts of the paper.

The following information was supplied regarding the deposition of related data:

Dryad http://dx.doi.org/10.5061/dryad.9fd75.

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
