# Peer review of "Variable effects of temperature on insect herbivory"

_PeerJ, doi:10.7717/peerj.376_

## Round 0.1 · original submission · Major Revisions

I agree with the comments of both reviewers. There is much merit in this work, but the manuscript will need some substantial revision before it is acceptable for publication. In particular, please revise the manuscript in response to: 1) the comment of reviewer 1 regarding the life stages of the insects studied, 2) the comments of both reviewers about clarifying the analytical methods, and 3) the comment of reviewer 1 about the validity of the findings. Following on this last point, I caution the authors from concluding that "effects of temperature on herbivory rates are unpredictable" (line 287, also see line 51 and the Abstract). More precisely, this study shows variation in this relationship (no single predictable relationship), but there are likely to be factors that do predict the variation in this relationship. They are simply not known at this time, and not revealed by this study.

Minor comments
Line 158: "consumption rate lie..." has a grammatical mistake.
Line 195: "then" should be "than."

Reviewer 1 ·

Basic reporting

Mostly good, although there are two confusing things that need to be fixed.

1. Consumption rates by herbivores are reported to be "mass corrected g" but there is no indication of how this calculation was done. My guess is that they calculated how much a 1-g herbivore would eat--in that case, consumption should be reported as g/g. And the authors should state explicitly somewhere what they did.

2. There is a disturbing fuzziness about what life stages were used. I assumed that they used only larvae. Figure 1, however, shows an adult beetle, and suggests that some of the herbivores were adults. Alerted to this, I went back through the paper and couldn't find any place where this was specified--this MUST go into the paper, and if they used a mix of larvae and adults they would need to partition the data this way too.

Experimental design

The statistical modeling of the results seem oddly long and complicated; but I'm OK with the actual content. It's true that performance curves have complicated, strange shapes, and that you need multi-parameter models to capture them.

The other weakness is that the number of replicates per insect-plant pair is quite variable. The authors point this out, but it nonetheless detracts from the overall power of the study.

Validity of the findings

My main comment focuses on how the authors connect their findings to conclusions about how warming climates may affect top-control of plants by insects. Their conclusion boils down to basically this: "there is a lot of variation in the shapes of insect thermal performance curves (for feeding), so much so that our expectation of higher consumption at higher temperatures was not met." They then use this conclusion to knock down more theoretical predictions (from O'Conner) that higher temperatures will stimulate insect feeding more so than it will stimulate plant growth, so that insects will have bigger impacts in warmer worlds.

My problem with this conclusion, as it is developed in the ms, is that it's not connected to local patterns of temperature variation in the environments from which the insects were taken. The experimental temperatures used were 20, 25, 30, and 35--and between 20 and 30 feeding really did rise for most of the herbivore-plant pairs (see Figure 2). The major fall came at 35C. So the relevant question becomes what is the mean temperature at their site in Maryland, and what is the *frequency distribution* of temperatures. If the temperatures are often between 20 - & 25, then climate warming of several degrees really would increase the community level consumption of plants. If the mean temperature is 30 (which I really doubt) and it rises a few degrees, then consumption rates could fall. When the authors characterize patterns of local temperature variation in the text, they say simply that the range of temperatures 20 - 35 is observed. Figure S1 provides a lot more detail, but it suggests that mean temperatures lie in the range 20 - 25 or even below.

So in this sense there's a real mismatch between what the authors found (Figure 2), what they conclude (climate effects hard to predict), and actual patterns of temperature variation at their site (Figure S1), which suggest that most insects most of the time are at the cold end of their temperature range (and that warming a few degrees would stimulate consumption).

My strong recommendation is that the authors obtain (or simulate using a package in R like lars-wg) hourly patterns of temperature variation over an entire summer, and to couple those estimates with the measured performance curves for each insect-plant pair. In such an approach, they could then simply add 1, 3, 5, etc. degrees to the ambient predicted temperature and ask, over the summer, how much difference does it make. Such an approach would provide a vastly more convincing framework for evaluating claims about the effects of climate change on consumption.

Reviewer 2 ·

Basic reporting

Overall, this study presents an impressive amount of data on herbivore consumption thermal reaction norms and resource quality. The authors find considerable interspecific variation in these thermal reaction norms, but rather surprisingly marginal impacts of resource quality. In general, the study appears to be well conducted and appropriately interpreted. However, I do have several questions and concerns about the analyses and presentation of the results.

Major comments:
The Introduction would benefit from a greater explanation and focus on the concept of the thermal reaction norm. Much of this ms rests on interpreting differences in thermal reaction norm shape--indeed the entire conclusion hinges upon the fact that many studies do not account for declines in consumption under excessively high temperatures--but it's never explicitly mentioned that there is a minimum for performance (consumption in this study) below which consumption is zero, rising to an optimum and declining to a maximum value beyond which consumption is zero. Also in the Intro, the expectation for the exponential increase in herbivory with increasing temperature is not well motivated; in particular, it is unclear whether this expectation derives from empirical results or theoretical predictions. The Discussion would seem to suggest a theoretical basis, and if so, this should be made clearer earlier on.

The authors make much of their community level approach, and indeed, this is an important aspect of the study. Yet, the analyses would seem to ignore the community aspect in that they assume statistical independence among species. Even in the absence of species-level phylogenetic hypotheses, higher-level phylogenetic relationships can still be incorporated into the analysis to account for this non-independence. Either this needs to be incorporated or the authors at the very least need to justify its omission.

Minor comments:
In the Results, it would be helpful to define how 'significance' is being assessed for the MCMC models; not all readers will be familiar with the Bayesian equivalent of a p-value.

Editorial comments:
Line 2 - 'import' should be 'important'

The text would benefit from greater clarity of the directionality of effects; for example, in the first sentence of the ms "Temperature can influence the top-down control of plant biomass by increasing herbivore metabolic demands.", directionality is given to 'increasing' herbivore metabolic demands, which should be matched with 'increasing' temperature, but is not currently presented as such.

Some of the transitions between sentences and paragraphs could also use some careful editing. For example, the second and third (and again in the sixth) sentences (in the abstract) each begin with 'however'.

Experimental design

Fine except for the issue of phylogenetic non-independence described above

Validity of the findings

Fine

---

## Round 0.2 · accepted · Accept

I commend the authors for their serious efforts to revise and improve the manuscript. I agree that it has become a stronger paper and one that makes an important contribution to understanding biotic effects of climate change.